# Friction and Wear Characteristics of Polydimethylsiloxane under Water-Based Lubrication Conditions

**DOI:** 10.3390/ma15093262

**Published:** 2022-05-02

**Authors:** Sung-Jun Lee, Yoon-Chul Sohn, Chang-Lae Kim

**Affiliations:** 1Department of Mechanical Engineering, Chosun University, Gwangju 61452, Korea; k3668591@chosun.kr; 2Department of Welding and Joining Science Engineering, Chosun University, Gwangju 61452, Korea; yoonchul.son@chosun.ac.kr

**Keywords:** friction, lubrication, PDMS, surfactant, wear

## Abstract

In this study, the friction and wear characteristics of polydimethylsiloxane (PDMS) were evaluated when using lubricants created by adding surfactants at various ratios to deionized (DI) water. When pure DI water is used as a lubricant, the repulsion of water from the hydrophobic PDMS surface is large and the interfacial affinity is low; thus, the lubrication properties cannot be significantly improved. However, when a lubricant with a surfactant is added to DI water, the interfacial affinity with the PDMS surface increases to form a lubricating film, and the friction coefficient is greatly reduced. In this study, under dry and pure DI water conditions, severe wear tracks were formed on the PDMS surface after 10,000 cycles of reciprocating sliding motion under a vertical load of 100 mN, whereas in the case of the surfactant-based and water-based lubricant, no severe wear tracks occurred. The friction and wear characteristics of the PDMS were evaluated by increasing the normal loads and sliding cycles with a water-based lubricant containing 1 wt % surfactant. Under normal loads of 300 mN and 500 mN, only minor scratches occurred on the PDMS surface up to 10,000 and 100,000 cycles, respectively, but after 300,000 cycles, very severe pit wear tracks occurred.

## 1. Introduction

Polydimethylsiloxane (PDMS) is a material with excellent moldability and is used in various fields, such as electrodes and displays [1,2,3,4]. In addition, studies have been conducted aiming to apply PDMS as a biomaterial, owing to its excellent biocompatibility [5,6]. However, PDMS becomes easily worn owing to its low mechanical properties and high friction properties. As such, research is needed to solve this problem [7].

PDMS is a thermosetting material fabricated by mixing a base and curing agent and applying heat to harden them. Because the mechanical properties differ depending on the curing conditions, it is easy to control the mechanical properties [8,9,10]. Typically, a 10:1 mixing ratio is used, but the mechanical properties can be improved by increasing the curing agent ratio. In addition, studies have been conducted to improve the mechanical properties of specimens by curing them at high temperatures for a long time [11,12]. However, if the curing agent mixing ratio exceeds a certain ratio, voids and non-uniformities occur owing to the excessive amount of curing agent, and the mechanical properties deteriorate [13,14]. Although the mechanical properties of PDMS can be improved by adjusting the curing agent ratio and curing conditions, PDMS nevertheless maintains high friction properties, owing to its high surface adhesion [12]. For this reason, studies have been conducted to improve the mechanical and tribological properties of PDMS by adding micro/nanoparticles [15,16,17]. The mechanical properties are improved by the micro/nanoparticles. Simultaneously, the high adhesion of the PDMS is removed, reducing the friction and abrasion and improving the durability. However, when it is necessary to use pure PDMS (i.e., without forming a composite), a lubricant must be used to improve the friction properties [12,18].

Compared to the dry condition, when a lubricant is applied between the surfaces of two contacting objects, the direct contact between the two contact surfaces is reduced by the lubricating film formed on the surface, thereby reducing friction and wear [19]. However, some normal lubricating oils are difficult to apply to the living body due to their chemical composition, and there is a problem that causes environmental pollution due to waste lubricants [20]. In order to solve these limitations, recent studies on water-based lubrication have been conducted [21,22]. Water is bio-friendly, environmentally friendly, easy to supply compared to lubricants, and has economic advantages, but it is not easily applied as a lubricant in various fields because it has low viscosity and corrodes the surface of metal. Recently, a study was reported to improve the lubrication properties by adding nanoparticles to water to improve the water lubrication properties [23]. 

As PDMS is commonly used as a biomaterial, there is a need to lower the high friction coefficient of PDMS using bio/environment-friendly lubricants. In this study, based on evaluations of the friction/wear characteristics, and the durability of PDMS under lubricating conditions, we propose a bio/environment-friendly lubricant suitable for PDMS. The friction and wear characteristics of PDMS are evaluated for a water-based lubricant with an added surfactant.

## 2. Materials and Methods

### 2.1. Materials

The PDMS specimen (Sylgard 184 base and curing agent, Sewang Hightech, Kimpo Korea) was prepared by mixing a PDMS base and curing agent at a weight ratio of 10:1. After mixing the PDMS base and curing agent in a 60 mm-diameter Petri dish, air bubbles were removed at room temperature for approximately 30 min. Then, after heating at 80 °C for 2 h in a heating oven, the mixture was cured at room temperature for more than 24 h. The thickness of the cured PDMS specimen was approximately 2 mm, and it was cut to a size of approximately 15 × 15 mm for the experiment.

To improve the lubrication properties of the PDMS specimens, bio-friendly/environmentally friendly water-based lubricants were prepared. In general, when deionized (DI) water is used as a lubricant, a surfactant (Silwet L-77, YUWON INTEC Ltd., Gunpo, Korea) is added to the DI water. This is because the interfacial affinity of pure DI water with the hydrophobic PDMS surface is low, and the lubricating properties correspondingly deteriorate. In this study, Trisiloxane-based Silwet L-77 was selected as the surfactant, and it was added to increase the interfacial affinity between the PDMS and water to form a lubricating film on the PDMS surface. As surfactants have both hydrophilic and lipophilic groups, they are mainly used to prepare mixed solutions or to modify surface properties [24,25]. In addition, Silwet L-77 is expected to exhibit sufficient lubrication properties, owing to its excellent properties for reducing surface tension [26]. In this study, after adding Silwet L-77 to DI water at ratios of 0.1, 1, and 10 wt %, respectively, the aqueous solutions were respectively mixed for 1 h using a magnetic stirrer and uniformly dispersed through ultrasonication for 10 min. Through this process, five different lubricants were prepared: pure DI water lubricant without surfactant (0 wt %), surfactant–water-based lubricants (0.1 wt %, 1 wt %, and 10 wt %), and pure surfactant lubricant without DI water (100 wt %).

### 2.2. Experiments

The chemical compositions of the PDMS and lubricants were analyzed using attenuated total reflectance Fourier transform infrared spectroscopy (ATR-FTIR, Thermo scientific, Nicolet 6700, Seoul, Korea). We attempted to understand the quantitative/qualitative characteristics of the chemical components through analysis of the unique spectra and respective peaks of the water-based lubricants according to the surfactant content. The wavenumber range was 400–4000 cm^−1^, and the spectra were analyzed with a resolution of 2 cm^−1^.

The interfacial affinities between the surface of the PDMS specimen and surfactant–water-based lubricants of different ratios were compared. The contact angle was measured by dropping pure DI water, DI water containing 0.1, 1, and 10 wt % of surfactant, and 100 wt % of pure surfactant onto PDMS specimens prepared under the same conditions and processes. The interfacial affinity between the PDMS surface and different lubricants was evaluated based on the contact angle, and the changes in the interfacial affinity with the PDMS according to the surfactant content were analyzed. After each lubricant was dropped three or more times onto the PDMS surface, the average value of the measured contact angles was obtained for comparison.

The mechanical properties of the PDMS specimens were measured by using a custom-built indentation tester. An indentation test is a technique for measuring the indentation depth according to an applied load by applying the load after the tip is in contact with the specimen. The degree of resistance of the specimen to indentation deformation is evaluated. The experimental conditions are listed in Table 1, and they are summarized as follows. A steel ball tip with a diameter of 25.4 mm was used, the loading/unloading speed was 0.3 mm/s, and a maximum load of 500 mN was applied. The change of the load according to the depth at which the ball tip was pressed during the loading process and change in the indentation depth according to the load removed during the unloading process were analyzed. To ensure reliability, the experiments were repeated three or more times under the same conditions, and all of the experiments were conducted under temperature and humidity conditions of approximately 25 °C and 30%, respectively.

A reciprocating-type tribotester (RFW 160, NEOPLUS Co., Ltd., Daejeon, Korea) was used to evaluate the lubricating properties of the PDMS. The tribotester used in this study employed a method in which the balance weight was moved while the specimen was fixed on the stage and aligned with the specimen, and then, a load was applied by increasing the weight. A 1 mm-diameter steel ball was used as the counter tip, and a 2 mm stroke was performed with a reciprocating sliding motion at a speed of 16 mm/s. The vertical load was increased to 100, 300, and 500 mN, respectively, and the number of reciprocating sliding repetitions was increased to 10,000, 100,000, and 300,000 cycles, respectively. The friction characteristics of PDMS under dry and lubricated conditions were also evaluated. The friction and wear characteristics were compared by applying lubricants with different surfactant addition ratios. The experimental conditions for the tribo-tests are summarized in Table 2. Each experiment was repeated three or more times under the same conditions to ensure reliability in the experimental results. After the tribotest, the morphologies of the wear tracks formed on the PDMS surface were analyzed using a 3D laser scanning confocal microscope (VK-X200, KEYENCE, Osaka, Japan).

## 3. Results and Discussion

Figure 1a shows the results from the ATR-FTIR analysis of the PDMS. The PDMS shows strong peaks at 2962 cm^−1^, 1257 cm^−1^, 1010 cm^−1^, and 787 cm^−1^. The peak of the 2962 cm^−1^ infrared band is characteristic of hydrocarbons, and it indicates C-H stretching [27]. The peak at 1257 cm^−1^ indicates bending vibration owing to the vibration of hydrocarbons. The peak at 1010 cm^−1^ indicates Si-O-Si stretching. The peak at 787 cm^−1^ indicates the result of the CH_3_ rocking and stretching in Si-CH_3_ [28]. PDMS consists of CH_3_ and Si, and it is cured by combining them. From the ATR analysis of the PDMS, it can be seen that the peaks caused by the stretching and bending vibrations of the hydrocarbons and silicon show very similar trends to those from previous studies [27,28,29].

Figure 1b shows the results from the ATR-FTIR analyses of the DI water and water-based lubricants with surfactants at different weight ratios. For the DI water, strong peaks are observed at 3266 cm^−1^ and 1636 cm^−1^. In addition, a broad peak is observed in the region of 3000–3700 cm^−1^. The broad peak near 3266 cm^−1^ is a result of the hydrogen crosslinking of water molecules and indicates O-H stretching, and the peak at 1636 cm^−1^ is owing to H-O-H scissor bending [30]. The weak peak at 2128 cm^−1^ is the result of a coupling of scissors bending [30]. Silwet L-77, a surfactant, is composed of flexible Si-O-Si, hydrophobic methyl, and hydrophilic polyoxyethylene [31]. The peak at 2866 cm^−1^ is owing to stretching caused by the bonding of the carbon and hydrogen [32]. The peak at 1349 cm^−1^ indicates CH_2_O deformation; it is owing to stretching and is showing a hydrophilic property of a trisiloxane [33,34]. The peak at 1000–1100 cm^−1^ results from the asymmetric stretching of Si-O-Si (1050 cm^−1^) and Si-O-CH_3_ (1100 cm^−1^), and the peak at 839 cm^−1^ indicates trimethyl [33,34]. In the case of the surfactant-added water, the peaks of DI water and surfactant are observed together, and no new peaks are observed. Although no chemical bonding is achieved, the peak near 1350 cm^−1^ is expected to show improved interfacial affinity with the PDMS surface, owing to the hydrophilic nature of the aqueous solution. 

As shown in Figure 2, the differences in the contact angles of the lubricants dropped on the PDMS surface were compared. In the case of pure DI water, as the PDMS surface has hydrophobicity, a large contact angle of approximately 108° is shown, whereas the surfactant–water-based lubricants show small contact angles of 20° or less. It can be seen that the interfacial affinity between the hydrophobic PDMS surface and DI water is improved by the addition of the surfactant. Among the cases where 0.1, 1, and 10 wt % of surfactants are added, the contact angle for 10 wt % is the smallest at approximately 13°, but there are no significant differences in the contact angles between them. In the case of 100% pure surfactant, the contact angle is approximately 23°; i.e., it is slightly larger than those of the surfactant–water-based lubricants. This is thought to be caused by the increase in cohesion between the molecules of the pure surfactant. That is, the surfactant increases the interfacial affinity, but it must be mixed with DI water in an appropriate ratio to be evenly applied on the hydrophobic PDMS surface. As the contact angle on the PDMS surface is significantly reduced relative to that of pure DI water even if only a small amount of surfactant is added (such as in the 0.1 wt % surfactant–water-based lubricant), it is expected that surfactant–water-based lubricants will be sufficient for use as lubricants for PDMS. 

The elastic modulus of the PDMS specimen was obtained using Equations (1)–(4) with the load-displacement data obtained from the indentation experiment on the PDMS specimen prepared with the PDMS base and curing agent in a mixing ratio of 10:1 [35].

In the equations shown below, *h_m_* is the indentation depth, *h_c_* is the contact depth, *dF/dh* is the stiffness, *F_m_* is the maximum load, *R* is the radius of the indenter, *A* is the real contact area, *E_r_* is the reduced elastic modulus, *E*_1_ is the elastic modulus of the specimen, *v*_1_ is the Poisson’s ratio of the specimen, *E*_2_ is the elastic modulus of the indenter, and *v*_2_ is the Poisson’s ratio of the indenter.
(1)$$h_{c}=h_{m}-\frac{3}{4\;}\frac{F_{m}}{\left(dF/dh\right)}$$
(2)$$A=\pi \left(2Rh_{c}-h_{c}^{2}\right)$$
(3)$$E_{r}=\frac{1}{2}\left(\frac{dF}{dh}\right)\sqrt{\frac{\pi }{A}}$$
(4)$$\frac{1}{E_{r}}=\frac{\left(1-v_{1}{}^{2}\right)}{E_{1}}+\frac{\left(1-v_{2}{}^{2}\right)}{E_{2}}$$

When a load of 500 mN is applied, approximately 0.3 mm is press-fitted, and the elastic modulus of the PDMS is calculated as 0.77 MPa. In previous studies, the elastic modulus of PDMS was reported to be approximately 0.5–1 MPa at a 10:1 mixing ratio; thus, the results of this study are considered to be reasonable [12,13,14,36]. The elastic moduli of polymer materials are lower than those of metal or ceramic materials. When a constant load is applied in the contact state, the contact area with the counter surface increases, owing to the increase in the indentation depth.

The friction and wear characteristics of the PDMS specimens were also evaluated under dry and lubricated conditions. The test in the dry condition was performed while the steel ball tip was in direct contact with the PDMS specimen without lubricant. The experiment under the lubricating condition was performed with five types of lubricants applied on the PDMS surface: pure DI water, DI water with 0.1, 1, and 10 wt % surfactants added, respectively, and 100% pure surfactant. After applying a vertical load of 100 mN, a reciprocating sliding motion of a 2 mm stroke was performed for 10,000 cycles at a sliding speed of 16 mm/s. The changes in friction coefficients and their average values were compared as shown in Figure 3. In the friction test under the dry condition, the friction coefficient of the PDMS maintained a high value of 1.0 or more through the initial 250 cycles of the sliding motion; then, the friction coefficient rapidly increased up to approximately 800 cycles, reaching approximately 1.85. It is thought that severe wear may occur on the PDMS surface in the section where the friction coefficient rapidly increases. As the surface of the PDMS becomes damaged, a wear track is generated, and the friction coefficient rapidly increases owing to the rough surface of the wear track. The maximum high-friction coefficient value of approximately 1.85 is maintained for a certain number of cycles, and then, it gradually decreases from approximately 1250 cycles to 10,000 cycles, converging to a friction coefficient value of approximately 1.7. The average friction coefficient for all 10,000 cycles in the dry condition is 1.7. When pure DI water is used as a lubricant, the average friction coefficient of the PDMS is 0.84, and no abrupt change in the friction coefficient is observed during the entire cycle. In the case of lubrication by DI water, the friction coefficient is reduced approximately twofold relative to the test in the dry condition, but the value remains high. At the beginning of the sliding motion, the friction coefficient appears to be less than 0.8, but it gradually increases over 300 cycles (reaching 0.86), converges to a value of 0.84, and then remains constant for the entire 10,000 cycles. It is thought that the friction coefficient increases as the surface of PDMS is scratched by the steel ball at the beginning of the sliding motion. As shown in Figure 2, when the contact angle of the pure DI water is as large as 108° and DI water is used as a lubricant, the interfacial affinity between the hydrophobic PDMS surface and DI water is low, suggesting that the DI water does not play a sufficient role as a lubricant [36]. It is considered that the lubricating film is not properly formed on the surface of the PDMS because of the large contact angle, and thus, the friction coefficient maintains a relatively high value. As shown in Figure 3, when an aqueous solution containing surfactant is used as a lubricant, the friction coefficient of the PDMS is 0.04–0.08; i.e., it is greatly reduced by 95–97% and 90–95% relative to the dry and pure DI water lubrication conditions, respectively. As shown in Figure 2, the surfactant used in this study reduces the surface tension, so even if a small amount is added to DI water, the contact angle is 20° or less; i.e., it is greatly reduced compared to that of pure DI water. In addition, despite the low viscosity, the interface affinity between the PDMS surface and aqueous solution improves enough to form a sufficient lubricating film, so the friction characteristics are improved. There are no significant differences between the friction coefficient values of the surfactant–water-based lubricants with 0.1, 1, and 10 wt % of surfactant added, respectively; the friction coefficient of the 1 wt % surfactant–water-based lubricant is the lowest at approximately 0.05. The 100 wt % surfactant shows a friction coefficient of approximately 0.04, i.e., the lowest value among lubricants, but the difference is very slight, and it is economically inefficient to use 100% surfactant as a lubricant. By comparing the changes in the friction coefficient according to the addition ratio of the surfactant, it can be confirmed that even if a small amount of surfactant (approximately 1 wt %) is added to DI water, friction characteristics similar to those of 100 wt % surfactant can be exhibited. Figure 4 shows images of the wear tracks formed on the PDMS surface from the sliding motions performed under dry and DI water lubrication conditions. Under the dry condition, when the PDMS surface is pressed by the counter tip, parts that rise above the surface are generated around the tip, which causes an increased contact area between the tip and the PDMS surface and a phenomenon that blocks the direction of the tip, resulting in high friction. Moreover, owing to the adhesive properties of PDMS, it easily adheres to the surface of the counter tip, and a significant amount of force is required to counter the adhesive force, resulting in increased friction [37]. Therefore, as this process is repeated, scratches and wear particles are generated on the PDMS surface, as shown in Figure 4a [7]. The frictional force increases rapidly owing to the damage generated on the surface, and the wear is accelerated by the rough surface [38]. In addition, as the wear particles adhere to the PDMS surface by the adhesive force of the PDMS, the wear shape forms as if it has risen above the surface. While severe wear tracks were generated on the PDMS surface by the experiments performed under dry conditions, when DI water is used as a lubricant, some deep scratches and wear particles were generated on the surface of the PDMS. From the fact that the friction coefficient in DI water lubrication condition was lower than that in dry condition, the lubrication effect of DI water was shown to some extent, which indicates that there was a difference in wear characteristics. However, owing to the high surface tension of the DI water, the lubricating film cannot properly form, and the counter tip and PDMS surface are in direct contact in many cases. This results in severe damage to the surface, as shown in Figure 4b. Moreover, a high frictional force is induced in the process of recovery of the compressed specimen owing to the failure to form a lubricating film. As a result, the PDMS is peeled off, causing the wear particles to fall off. When DI water is used as a lubricant for PDMS, the adhesive force can be reduced to some extent, but the friction and wear characteristics are not significantly improved, because the lubricating film is still not properly formed. When DI water with an added surfactant is used as a lubricant, almost no wear occurs during a friction test of 10,000 cycles. The surface tension of the DI water is reduced by the surfactant, and the interfacial affinity with the PDMS increases. During the 10,000 cycles, a sufficient lubricating film is formed on the surface of the PDMS and significantly reduces the direct contact between the counter tip and PDMS. Thus, a low friction coefficient is maintained, which is considered to be the result of nearly zero abrasion on the surface of the PDMS.

The friction and wear characteristics of the PDMS vary according to experimental conditions such as the vertical load, sliding distance, sliding speed, temperature, and humidity [7,39]. To confirm the lubrication conditions for protecting the surface of PDMS, the experimental conditions were harsh, with a high load and long time. The friction coefficient of the 1 wt % surfactant–water lubricant was measured as the lowest among the surfactant–water-based lubricants except for the 100 wt % surfactant, and this was used as the lubricant to proceed with sliding under high-load and long-cycle conditions. Figure 5 compares the friction coefficients according to the load for the sliding motion during 10,000 cycles. The friction coefficients under the load conditions of 300 and 500 mN increase compared to the load condition of 100 mN. In the high-load condition, the press-fitting depth of the counter tip becomes deeper than that in the low-load condition, and the contact area increases. This is thought to increase the energy loss of the PDMS and the friction coefficient [12]. As shown in Figure 5a, both 300 and 500 mN load conditions show a friction coefficient of 0.1 or more at the beginning of the sliding motion, but it immediately decrease to below 0.1 as the lubrication properties are improved by the 1 wt % surfactant–water lubricant. Moreover, the low friction coefficient is maintained for 10,000 cycles. It is judged that the 1 wt % surfactant–water lubricant sufficiently forms a lubricating film and maintains a low friction coefficient even when the press-fitting depth increased owing to an increase in the vertical load. As shown in Figure 5b, the average friction coefficients are 0.055, 0.086, and 0.072 under normal load conditions of 100, 300, and 500 mN, respectively.

Figure 6 shows wear track images of the PDMS according to the vertical load for 10,000 cycles of sliding under the condition of the 1 wt % surfactant–water lubricant. As shown in Figure 6a, for a normal load of 100 mN, no wear tracks or minor scratches are observed by a 3D laser scanning confocal microscope with a resolution of several tens of nanometers in the area where the sliding motion on the PDMS surface occurs. The 1 wt % surfactant–water lubricant maintains a sufficient lubricating film for a load of 100 mN, and it protects the PDMS surface without allowing any serious damage for at least 10,000 cycles. Figure 6b shows the wear track formed on the PDMS surface after sliding for 10,000 cycles with a vertical load of 300 mN. It can be seen that under a vertical load of 100 mN, a slight level of wear occurs relative to the cases (Figure 4) in the dry condition and when the DI water is used for lubrication. It is considered that the surface tension is reduced by the 1 wt % surfactant, and a sufficient lubricating film forms on the surface of PDMS, resulting in a lubrication effect superior to that of DI water. As shown in Figure 6c, the wear track formed on the PDMS surface from sliding for 10,000 cycles under the 500 mN load condition appears to be slightly intensified compared to the 300 mN case, but the 500 mN load condition represents a fivefold vertical load increase relative to the dry and DI water lubrication conditions (100 mN). Despite this, it can be seen that only a slight level of wear occurs. A lubricating film is evenly formed on the PDMS surface from the 1 wt % surfactant–water lubricant, reducing the direct contact with the counter tip and leading to a decrease in the frictional force. The wear of the PDMS surface is also reduced owing to the reduced direct contact and reduced friction. However, as the vertical load increases, the counter tip and PDMS surface come into direct contact in many cases, thereby increasing the frictional force between the two contact surfaces. Owing to the increase in the frequency of the direct contact and frictional force, relatively minor scratches are generated on the PDMS surface.

Figure 7 shows the wear track images of the PDMS when the sliding cycle is greatly increased under a high load of 500 mN and 1 wt % surfactant–water lubricant is used. Figure 7a,b show the wear traces formed on the PDMS surface after sliding motions of 100,000 and 300,000 cycles, respectively. It is judged that the 1 wt % surfactant–water lubricant maintains the friction and wear characteristics of PDMS well, as only slight scratches are generated on the surface of the PDMS even after 100,000 cycles, i.e., 10 times more than those of the previous experiments. However, when the sliding motion is further conducted for 300,000 cycles, a wear track deeply dented in the PDMS surface appears. It is considered that the 1 wt % surfactant–water lubricant is lost to the surroundings, owing to the dynamically moving reciprocating motion during the 300,000 cycles of the sliding motion; moreover, there is a possibility that the viscosity of the lubricant is lowered by the frictional heat. Accordingly, it is considered that a relatively serious wear track forms as the sliding cycle lengthens and the direct contact between the counter tip and PDMS surface increases.

## 4. Conclusions

In this study, we attempted to establish conditions for lubricants to improve the friction and wear characteristics of PDMS. To find a lubricant suitable for PDMS, a biocompatible material, biocompatible and environmentally friendly water, was used as the lubricant. However, PDMS has a low lubricating effect when lubricated with water alone owing to its hydrophobic properties, and it does not significantly improve the friction and wear characteristics. Therefore, a surfactant was added to apply water lubrication to the hydrophobic PDMS. In friction testing under a dry condition, the friction coefficient was measured as high owing to the high adhesion and poor mechanical properties of the PDMS. Moreover, the friction coefficient increased rapidly after the initial 250 cycles. When DI water was used as a lubricant, the friction coefficient decreased; however, wear still occurred because an even lubricating film was not formed. In the case of using surfactant-added water as a lubricant, no wear occurred in the test of 10,000 cycles under a load of 100 mN. However, wear occurred in high-load and long-cycle tests, and as the number of sliding cycles increased, the wear became more severe. The results from this study can be used to improve the friction and wear characteristics of PDMS materials with hydrophobic properties, and they can suggest the potential of applying eco-friendly lubricants. In the future, it can be used to contribute to the development of eco-friendly lubricants through additional research, such as on the addition of nanoparticles. In addition, the results from this study can be used as basic data for research on the development of eco-friendly lubricants.

## Figures and Tables

**Figure 1 materials-15-03262-f001:**
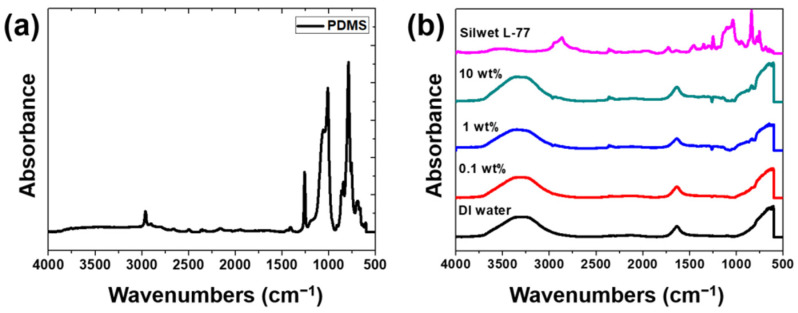
FTIR-ATR results of (**a**) PDMS and (**b**) lubricants.

**Figure 2 materials-15-03262-f002:**
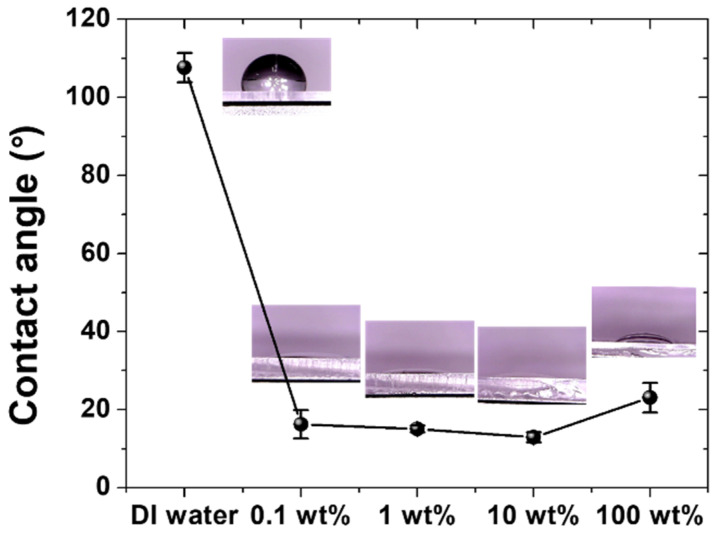
Contact angles of lubricants.

**Figure 3 materials-15-03262-f003:**
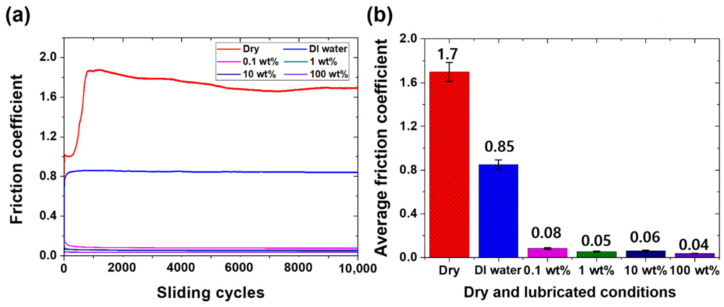
Friction characteristics of PDMS specimen under 100 mN normal load, dry, and lubricated conditions during 10,000 cycles. (**a**) Variation values of friction coefficient with respect to number of sliding cycles and (**b**) average values of friction coefficient.

**Figure 4 materials-15-03262-f004:**
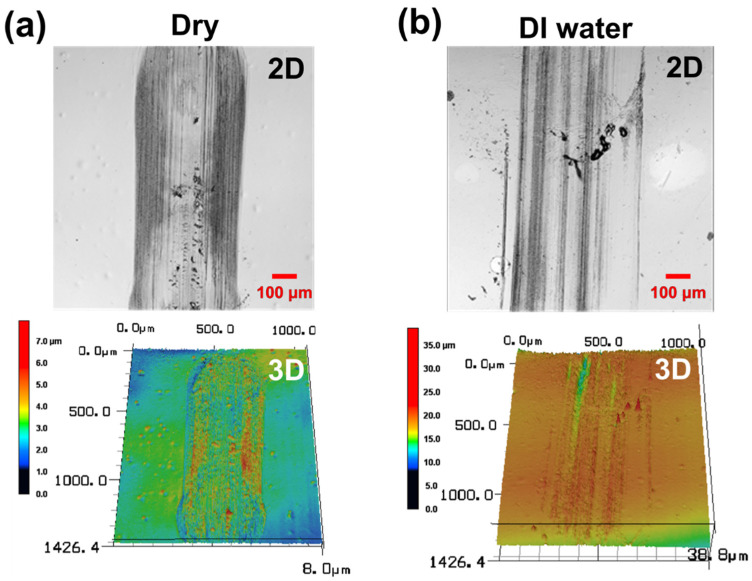
Confocal microscope 2D/3D profile images of wear track of PDMS specimen under 100 mN normal load after 10,000 cycles. (**a**) dry and (**b**) DI water lubricated conditions.

**Figure 5 materials-15-03262-f005:**
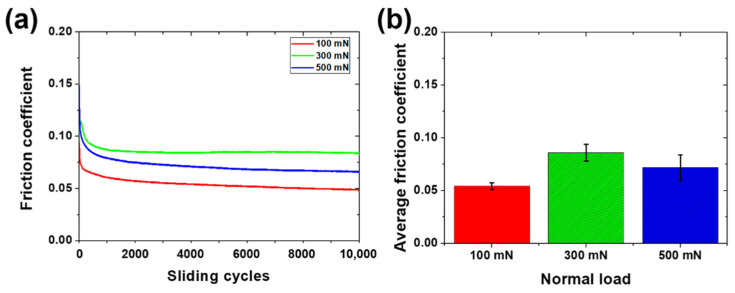
Friction characteristics of PDMS specimen under 100 mN, 300 mN, and 500 mN normal loads and 1 wt % surfactant–water lubricated condition during 10,000 cycles. (**a**) Variation values of friction coefficient with respect to number of sliding cycles, (**b**) Average values of friction coefficient.

**Figure 6 materials-15-03262-f006:**
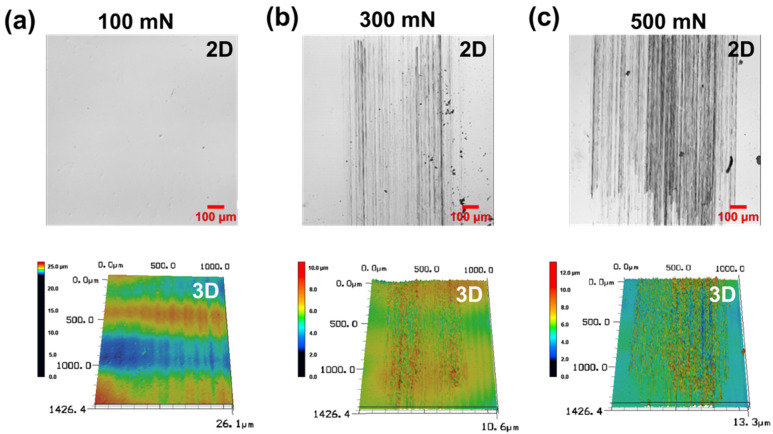
Confocal microscope 2D/3D profile images of wear track of PDMS specimen under (**a**) 100 mN, (**b**) 300 mN, (**c**) 500 mN normal loads and 1 wt % surfactant–water lubricated condition after 10,000 cycles.

**Figure 7 materials-15-03262-f007:**
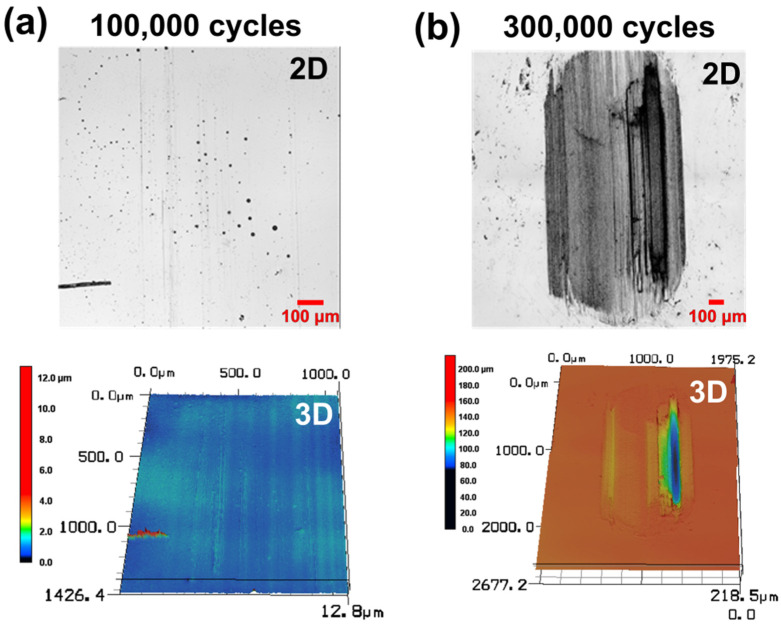
Confocal microscope 2D/3D profile images of wear track of PDMS specimen under 500 mN normal load and 1 wt % surfactant–water lubricated condition after (**a**) 100,000 and (**b**) 300,000 cycles.

**Table 1 materials-15-03262-t001:** Indentation test conditions.

Indentation Test	
Tip	Steel ball (D: 25.4 mm)
Loading/Unloading speed	0.3 mm/s
Max. load	~500 mN

**Table 2 materials-15-03262-t002:** Tribo-test conditions.

Tribo-Test (Reciprocating Type)	
Tip material (Diameter)	Steel ball (D: 1 mm)
Normal load	100/300/500 mN
Sliding speed	16 mm/s
Sliding stroke	2 mm
Sliding cycle	10,000/100,000/300,000 cycles
Lubrication condition	Dry/DI water/0.1, 1, 10, 100 wt % Surfactant–water-based lubricants

## Data Availability

Data are available on request from the corresponding author.

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
