# Peer review of "Friction and Wear Characteristics of Polydimethylsiloxane under Water-Based Lubrication Conditions"

_materials, 2022, doi:10.3390/ma15093262_

Round 1

Reviewer 1 Report

comment on article :

Some sentences in the manuscript is too long and must be improved. Some sentences is not very comprehensible and must be improved. For exemple page 6 line 198-200, page 7 line 243-246.

improve quality of color bar of figures 4, 6 and 7. Could authors used the same level of color bar of this figure. This will improve the comprehension of the figures and make it possible to better see the extent of the wear. In fact, in FIG. 7 for example, the blue in the figure on the left corresponds to a depth of 1 μm, whereas in the figure on the right it corresponds to 80 μm. With the same scale, we will immediately see that the figure on the right shows greater wear.

In table 1. Max. load is 50 mN whereas on the text (line 115 page 3) maxiumu load is 500mN. Which value is good ?

Equation 1 to 4: could authors put reference of article in which this formula is derived.  

Figure 3: level of friction for 0.1 1, 10 and 100%wt is too low to be seen. Could authors put mean value of friction coefficient on the figure?

Explanation of wear mechanisms is not cleaur on page 7 line 254-257.

Authors say“When DI water is 261 used as a lubricant, relatively less wear of the PDMS occurs,” but on figure 4 it seems that wear is more importante in DI water lubricated conditions?

Major REvision

Reviewer 2 Report

The appropriate file is attached.

Reviewer 3 Report

This paper aims to discuss the friction and wear of PDMS when using a lubricant compared with the dry condition.

The article is well organized and each section is well developed, showing the relevant results with an adequate discussion of them.

Although the paper is well written, some parts of the text must be revisited and rewritten to clarify the information to be passed for the reader. I suggest to review the entire paragraph that starts at line 49 to clarify the text.

In this way, I believe that the paper should be accepted for publishing with minor revisions in MDPI Materials.
